# Influence of Circular Economy Phenomenon to Fulfil Global Sustainable Development Goal: Perspective from Bangladesh

**Muhammad Azizuddin [1], Ahm Shamsuzzoha [2],[*] and Sujan Piya [3]**

1   Faculty of Business and Law, School of Strategy and Leadership, Coventry University, Coventry CV1 5FB, UK; ad7655@coventry.ac.uk
2   Digital Economy Research Platform, School of Technology and Innovations, University of Vaasa, P.O. Box 700, FI-65101 Vaasa, Finland
3   Department of Mechanical and Industrial Engineering, College of Engineering, Sultan Qaboos University, P.O. Box 50, Muscat PC 123, Oman; sujan@squ.edu.om
*   Correspondence: ahsh@uwasa.fi

**Abstract:** This paper highlights the extent of the relationships between circular economy (CE) practices and the implementation of the United Nations Sustainable Development Goals (SDGs). Specifically, the paper takes part in academic debates regarding CE and SDGs. It qualitatively investigates national governments' policy response and practices, with a focus on Bangladesh. The study finds varying degrees of momentum in the national policy response to SDGs and thus, it answers two research questions: (i) what is the relevance of CE practices to the United Nations (UN) SDGs? (ii) What are the responses from the Bangladeshi government to fulfil the UN SDGs regarding sustainable consumption and production with CE? As CE is a global trend, the research suggests that broad, conscientious connection and collaboration at the national level are essential. The findings implicate national governments in developing countries and UN SDGs for their policies and programme reassessment, considering the impact of the COVID-19 pandemic on sustainable development.

**Keywords:** sustainability; circular economy; sustainable development; COVID-19; SDGs; Bangladesh

## 1. Introduction

The concept of circular economy (CE) can be defined as the practice to manage resource circularity, efficiency and optimisation that proposes to use wastes as resources to create value [1,2]. It contributes to enhancing productivity and improving efficiency in managing both natural and human resources [3,4]. It has gained increasing attention globally in order to achieve local, national and global sustainability aims [5–9]. However, such attention is mostly from developed countries and rarely from developing countries (e.g., low- and middle-income countries), except China [10–12]. The aim of CE is essentially to create a balance among the economy, environment and society in the form of extending products' lifecycle or by returning products and material leftovers in the system to be reused [13–15]. Moreover, it contributes to achieving the Sustainable Development Goals (SDGs) in developing countries [16]. Conversely, SDGs also promote CE practices.

CE has been used as a strategy for solving sustainable development challenges and to bring about a positive impact on the environment, society and economy [17–20]. The activities related to CE generate momentum in life through the involvement of people in economic activities. Thus, CE helps the ultimate goals of poverty eradication, reducing environmental devastation and increasing the generation of new added value [21]. The United Nations (UN), the European Union (EU) and countries worldwide as a whole have emphasised natural resource conservation, and the concept of CE supports multiple international development agendas on energy, economic growth, sustainable cities, sustainable consumption and production, climate change, oceans and life on land [22–24]. Governments around the globe, including in Bangladesh, have considered CE as a "means

to ends" to accelerate the implementation of the UN's development agenda for 2030. Governments have taken sustainable consumption and production strategies to comply with CE principles, in line with the UN's SDGs. They have also given necessary directions through various legislations to ensure the link between CE practices and UN SDGs.

Even though a plethora of research has been carried out to look at the relationship of CE with SDGs in developed nations, it is still lacking in developing countries. The academic literature on sustainable development indicates that CE is a new emerging business model that is replacing the traditional linear business model and can be used as a useful tool for the solution of contemporary global sustainable development challenges [25]. This study is an endeavour to look into the issue with reference to a developing country, Bangladesh, and its government to represent the tip of the iceberg that contributes to filling in the academic gap in sustainable development literature. In addition, this study also discusses how policymakers and development practitioners worldwide could adopt required strategies to deal with sustainable practices with respect to CE. Thus, it paves the way for further in-depth research in the field.

Based on the stated circumstances as discussed above, this paper participates in academic debates on the concept of CE with the UN's SDGs. The objectives of this study are to investigate the relationship between CE and the UN's SDGs and to qualitatively analyse national governments' policy responses and partnerships globally, with a focus on Bangladesh. Based on the stated objectives, the study aims to answer the following two research questions (RQs):

- RQ 1: What is the relevance of circular economy practices to the UN's Sustainable Development Goals?
- RQ 2: What are the responses from the Bangladeshi government to fulfil the UN's SDGs regarding sustainable consumption and production with CE?

The rest of the article is organised as follows. Section 2 presents the literature review related to defining the key concepts of sustainability, sustainable development and circular economy, and highlights a theoretical framework that extensively displays the relationship between SDGs and CE. The study methodology is illustrated in Section 3, while Sections 4 and 5 briefly discusses the status of CE practices and SDG initiatives in Bangladesh, respectively. The overall study findings are explained in Section 6 with respect to policy and execution status of CE and SDGs in Bangladesh. Finally, Section 7 concludes the study, providing a constructive summary of the discussion that includes recommendations and implications for policy and execution issues with future research directions.

## 2. Literature Review

This section covers the various aspects of sustainability and sustainable development with respect to CE strategies and practices. In addition, it highlights the overall status of selected SDGs and their present global trends. Furthermore, a conceptual framework for CE with respect to SDGs is elaborated with the aim of understanding the impact of CE practices on the fulfilment of relevant SDGs.

### 2.1. Sustainable Development through CE Strategies and Practices

Sustainability is a word and science that is usually associated with development and denotes various connotations such as "living within means", "balance between spaces", "responsible consumption", "ability to exist constantly", etc. The awareness of sustainability is increasing in society, which can impact environmental, economic and social dimensions of SDGs. In terms of the environmental aspect, sustainability offers a reduction in emissions and waste, while regarding the economic aspect, it contributes to creating new opportunities for organisations through new regulations. From the societal perspective, sustainability creates the opportunity for a sharing economy [26]. Generally, it means the capacity for the biosphere and human civilisation to co-exist and focuses on meeting the needs of the present without compromising the ability of future generations [27]. With the issues of development, sustainability is a central concept of discussion. The academic debates and

practices in this domain are mixed, partly because of the sustainability dimensions, which are catalogued and somewhat unequally addressed [28]. However, it is parallel to enduring socio-economic development.

In 1987, the World Commission on Environment and Development (WCED) defined sustainability for the first time as "development that meets the needs of the present without compromising the ability of future generations to meet their own needs" [29–32]. It promotes building towards an inclusive, resilient and sustainable future for people and planet through fighting against poverty [33,34]. It is also considered as an umbrella concept that incorporates "development" in its approach, methods and techniques. There is a functional relationship between sustainability, business and development that is a crucial part of the CE. As a new socio-economic and business phenomenon, CE is a business model [35] that focuses on recycling, reduction and re-use. It involves the shift of existing reserves to renewable energy sources; this then creates economic, natural and social capital, and an environmentally friendly atmosphere for the development of a just society [36].

The CE concept is opposite of a "take-make-use-dispose" pattern of growth and is based on the 6R principles of reuse, recycle, redesign, remanufacture, reduce and recover [21,37]. This concept is based on the closed loop principle of a natural ecosystem, where there ideally exists no waste output; all input and waste output enter the circle of the ecosystem that essentially extends the life cycle of products [38]. According to the Ellen MacArthur Foundation (EMF), the CE is an "industrial system that is regenerative and restorative by design, rethinks products and services to design out waste, and negative impacts and builds economic, social and natural capital" [39]. It emphasises that there is a recognition of the economy needing to work efficiently at all levels—both locally and globally for individuals, organisations and businesses. The EMF states that "a circular economy aims to redefine growth, focussing on positive society-wide benefits and entails gradually decoupling economic activity from the consumption of finite resources, and designing waste out of the system" [39].

According to Bocken et al. [20], CE advocates systems of closing, slowing and narrowing the loop. Closing the loop includes cradle-to-cradle material, through a process of recycling, reuse, remanufacture and maintenance. This is to prevent entering the disposal stage [40]. Narrowing the loop would adopt fewer resources in a product with higher efficiency. An example of a closing and narrowing the loop is Evian plastic bottles [41]. FFC Information Solution Private Limited (2020) stated that Evian had removed the plastic label on their new product design and manufactured a new bottle with recycled bottle material [42]. Therefore, the material loop is narrowed and closed. Reducing material flow serves to lengthen the product life duration via product enhancement [43]. Apple Inc., California, USA, is a case in point, which is constantly developing robust materials, such as screens and batteries, to increase its product life from usage to disposal [43].

The current economic situation and activities, both in terms of capacity and format, pose a serious threat to sustainability [44–46]. They are traditionally based on the "take-make-use-dispose" pattern with wider consumption of natural resources. Therefore, the transition from a linear economy to a CE is the need of the hour. From a business perspective, the transition to a CE has significant impact on economic growth in the global economies [47,48]. With respect to economic reforms, the transition of CE also influences environmental and research policies in organisations. Ramani [37] claims that the "Persistent deterioration of natural resources, greater contamination of air, water and soil, diminishing biodiversity, emergence of new types of pathogens, climate change and heightened fragility of human health (even when longevity is increased) are being noted". These phenomena will have serious impact on sustainability goals.

Empirical studies by EMF [39] assert that CE designs, innovative business models, reverse cycles and enabling conditions are the essential building blocks for the transition to CE. The areas of circular design include material selection, standardised components, designed-to-last products, design for easy end-of-life sorting, separation or reuse of products and materials and design-for-manufacturing criteria that consider possible useful



applications of by-products and wastes. Innovative business models are always profitable, and initiatives will inspire other players and will be copied and expanded geographically.

Reverse cycles are new, requiring additional skills for the material decomposition and back into the industrial production system. This includes delivery chain logistics, sorting, warehousing, risk management, power generation and even molecular biology and polymer chemistry [40]. With improved collection and treatment of wastes, and more robust segmentation of end-of-life products, the leakage of materials out of the system will decrease. This, in turn, would promote the economics of circular design. In addition, market mechanisms need to play a prominent role to introduce CE principles, reinforced by policymakers, educational institutions and popular opinion leaders, for the widespread reuse of materials and higher resource productivity to become more commonplace [1,49–51]. Other factors such as collaboration, reviewing enticements, creating and implementing an appropriate set of international environmental rules, driving upscale fast and access to financing could further improve CE principles.

*2.2. SDGs Status and Current Global Trends*

The review of debates, documents and reports of SDGs so far generally indicates the gulf of difference between the set targets and achievements and the trend is far from the development path [52]. As we can see, economic growth and prosperity have depleted the world's natural resources and environment at an unprecedented speed. All around us, we can observe the consequences of climate change [30]. It is essential to push back these trends by transforming the current development path. Various discussions and reports of SDGs have provided ample information to countries to create a new development path [53]. Some recommendations are worth mentioning here, such as leave no one behind, transform economics for jobs, build peace and accountable institutions, focus on sustainability—environmental, economic and socio-political—and try to forge new global partnerships, free from conflicts and tensions [27]. The themes have prioritised marginalised and underrepresented groups to provide a level playing field.

On a national level, each country needs to create its own development path, balancing both the SDG philosophy and the specific constraints and potential of the nation. In comparison with the Millennium Development Goals (MDGs), the SDGs provide greater autonomy and flexibility for national governments. More specifically, the SDGs provide a valuable opportunity for the countries to shift towards an inclusive, fair and sustainable development path. The UN's list of sustainable goals and targets help to highlight the status of the SDGs and track progress.

*2.3. Conceptual Framework of Circular Economy for SDGs*

The SDGs are a global development programme of the UN's blueprint to achieve a better and more sustainable future for all, containing 17 goals adopted in 2015. They address the challenges that global populations are facing, including poverty, inequality, climate change, environmental degradation, peace and justice [5]. The programmes generally emphasise sustainable economic growth and strengthening modes of sustainability to implement the 2030 agenda for sustainable development [54]. Of these SDGs, SDGS 7, 8, 9, 11, 12 and 17 are, directly and/or indirectly, linked to the phenomena of sustainability and circularity. Governments in the countries that are signatories of the UN's SDGs play a crucial role in the target achievement by taking different actions, including national policies and programmes, related to the SDGs.

In order to achieve the SDGs targets, governments must initiate and deliver various economic activities as a means of promoting the sustainability approach [55]. Specifically, there are shared premises among multiple SDGs: 7 on energy, 8 on economic growth, 11 on sustainable cities, 12 on sustainable consumption and production, 13 on climate change, 14 on oceans and 15 on life on land that links to CE. According to General Assembly and ECOSOC Joint Meeting [53], this is a system of economy in which waste and pollution do not exist by designing and producing products and services and consumption of resources.

This is argued to be considered as compulsory as it could reduce global emissions by 3.6 billion tons per year by 2030. Thomson argues that "embedding the principles and practices of the circular economy into consumption and production regimes will be the key transition for achieving the SDGs" [56].

Global, regional and local legislation such as policies, rules and regulations and procedures and operational guidelines have a profound influence on SDG target achievements and the realisation of goals. The EU's Circular Economy Action Plan and the European Circular Economy Stakeholder Platform at a regional level, Nigeria's Extended Producer Responsibility (EPR) operational guidelines at a national level and the EU's MoU with China at an international level are cases in point. These support progress towards the 2030 agenda. The importance of business interest and stakeholder activism for a CE cannot be undermined. Daly reports in her study [57] focussing on the US that "the transition to a circular economy will be driven by business interest rather than national regulations".

Based on the above discussions, it is clear that there is a tight relationship between CE and the implementation plan for SDGs. This relationship can be conceptualised through a framework as proposed in Figure 1. From the figure, it is noticed that there are clear interrelationships or interdependencies between CE and SDG target achievements. It is also visible that SDGs can be achieved through CE by adopting/following several basic principles such as government policies and guidelines, interaction/partnership between government and private stakeholders and mind-set for circular business interests. From a policy perspective, the government of an individual country must propose and implement necessary rules and regulations to achieve SDG targets. Moreover, there must be tight interactions or coordination between government stakeholders and private organisations with respect to policy implementation to achieve SDGs. Furthermore, to successfully achieve SGDs, there must be an established mindset that supports the circularity phenomenon among business enterprises.

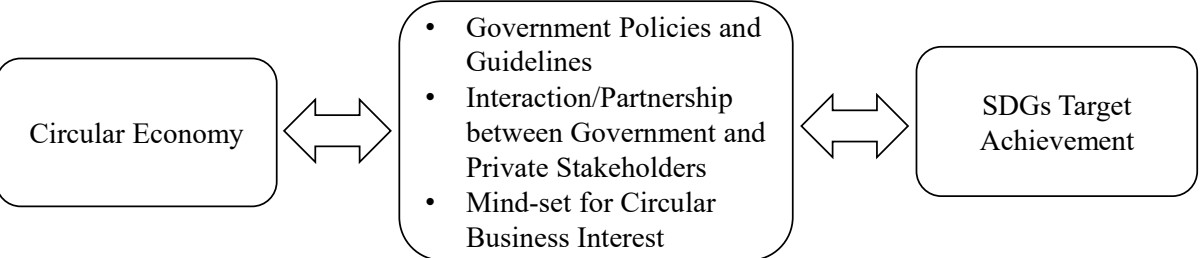

**Figure 1.** Conceptual framework of CE for SDGs.

## 3. Study Methodology

This research study adopted a qualitative approach in order to address two identified research questions. The qualitative method was divided into two separate sections: an extensive literature review and a Delphi approach in the form of semi-structured interviews, which was based on predefined questionnaires. It goes through the context and literature, and determines its underlying factors as a deductive process from broad to narrow scope [58]. The philosophical basis of this study is positivism under the paradigm of epistemology as an objectivist.

### 3.1. Research Agenda

The research agenda used in this study is to find the relationships between CE phenomena to fulfil global SDGs. Moreover, the main aim is to define the most popular methodologies that can be deployed to understand the impact of CE principles over SDGs. The keywords used for finding articles, books and documents associated to the research agenda are: "CE principles and strategies", "UN's SDGs", "government policy related to CE and SDGs", "circular production and consumption" and "sustainable environment".

These keywords work to identify the articles that are most likely studies related to the CE strategy.

### 3.2. Literature Search Criteria

In order to look for the relevant articles, the literature search was focused on peer reviewed journal articles, conference proceedings, published books, government policy documents and reports, the Bangladeshi government's various initiatives on CE and SDGs and other related publications. Several well-known major multi-purpose databases such as Web of Science, ProQuest, Emerald, Science Direct and EBSCO were used during this literature search. In addition, same keywords as discussed in Section 3.1 were used to look for more articles from Internet sources such as Google Scholar to increase the amount of useful literature in the field of interest. This search included work published up to September 2021.

### 3.3. Delphi Approach

Six experts in the field of sustainability, CE and SGDs were interviewed with pre-designed questionnaires (Appendix A) to understand their opinions related to the issues. As shown in Table 1, these experts included a policy maker, academic and researcher, development practitioner, international development expert, business manager, and a root level CE businessperson in Bangladesh. Other important information pertaining to the experts are shown in the table. They were selected due to their characteristics that share a common interest with the purpose of the study. We applied a purposive sampling technique to ensure a population with knowledgeable experts [59–63]. In fact, they are also familiar with the authors in a capacity of both academic and professional standing. The experts were interviewed during the period of December 2020 to March 2021 via telephone and WhatsApp calls. We discussed with the experts the issues of policy response to SDG realisation relating to CE practice in Bangladesh with open-ended questions. Their responses were spontaneous with elaborated discussions.

**Table 1.** Relevant information related to six experts.

| Expert Category | No | Affiliation | Experience (Years) |
|---|---|---|---|
| Policy Expert | 1 | Government | 20 |
| Academic/Researcher | 1 | University | 25 |
| Development Practitioner | 1 | NGO | 15 |
| International Development Expert | 1 | Intl. Dev. Organisation | 18 |
| Business Manager | 1 | Multinational Business | 14 |
| Circular Economy Business Person | 1 | Local CE Business Owner | 10 |

## 4. Status of Circular Economy Practices in Bangladesh

Bangladesh is a traditional, low-income, developing country in South Asia with a densely housed population. The country is an active player in the green global fora and mobilises resources to deal with climate change, environmental degradation and circularity [64–67]. The Bangladeshi government has taken a number of initiatives to make the country greener from the beginning of the 21st century. These initiatives are reflected in many strategy documents, laws and regulations, investment programs and the active participation in global efforts for a greener Bangladesh [68]. The Eighth Five Year Plan (July 2020–June 2025), the Perspective Plan 2041 and the Delta Plan BDP2100 have sought to integrate a greener Bangladesh in national planning.

There is a congenial policy environment in Bangladesh to promote CE that serves SDG achievement to implement the SDG agenda. This includes policy guidelines that scaffold a legal framework to think globally and act locally. Accordingly, the Bangladeshi government has introduced a mix of policy instruments in theory, such as fiscal incentives and mandates,

which may provide the signals for change; however, they are rarely operative in practice. Policy instruments such as the setting of portfolio standards, blending mandates for financing, mandatory performance standards and labelling, and green public procurement guidelines can be used to build new business opportunities and create markets for low-carbon, recycled or reused products and technologies [69].

The major policies and legal frameworks of the Bangladeshi government that are necessary to implement CE practices and fulfil the UN's SDGs were studied and are highlighted in Table 2. All the policies and legislative arrangements as shown in the table provide necessary guidelines related to CE that pave the way for the implementation of SDG agendas for sustainable development in Bangladesh [70].

**Table 2.** The Bangladeshi government's policies and frameworks to address CE and SDGs.

| Serial No. | Name of the Policy | Policy Year |
|:---:|:---:|:---:|
| 1 | Draft National Environmental Policy | 2018 |
| 2 | Solid Waste Management Rules | 2010 |
| 3 | National 3R Strategy for Waste Management | 2010 |
| 4 | Draft Solid Waste Management Rules | 2018 |
| 5 | Draft E-Waste Management Rules | 2018 |
| 6 | Draft SRO on Plastic Waste Management | 2019 |
| 7 | Renewable Energy Policy | 2008 |
| 8 | National Sustainable Development Strategy | 2008 |
| 9 | Bangladesh Industrial Policy | 2016 |
| 10 | Compulsory Use of Jute Package Act | 2010, 2013 |
| 11 | City Corporation Act (2009) | |
| 12 | Bangladesh Environment Conservation Act (with amendments) | 1995 |
| 13 | Bangladesh Environment Conservation Rules (with amendments) | 1997 |

## 5. Status of Sustainable Development Goals Initiatives in Bangladesh

In Bangladesh, several initiatives have been taken to fulfil the UN's sustainable development goals, especially related to the CE concept. Such initiatives are taken in various sectors such as energy, forestry, construction, garments, agriculture, fisheries, etc. In the energy sector, various renewable energy sources are used and studied to ensure sustainable energy production and distribution in Bangladesh [66–69]. In the forestry sector, several studies are conducted to uphold SGDs in Bangladesh [70–73]. Islam et al. [74] and Islam and Shamsuddoha [53] studied marine fisheries in Bangladesh with a view of sustainable conservation and management of coastal and marine fisheries industries.

Moreover, with energy-efficient manufacturing, and moving to a low-carbon CE, the construction industry is under pressure to reduce their embedded carbon whilst delivering more resilient infrastructure. The construction industries in Bangladesh practice sustainable energy use, lean construction strategy through using recyclable and low-cost materials to contribute to SDGs to the social, economic and cultural context of Bangladesh [75–77]. For instance, in the Bangladeshi construction industry, alternative cost effective building materials such as compressed stabilised earth blocks (CSEB) were produced using local soil and three different stabilisers (lime, cement and jute–lime mixture) at various percentages to withstand large deformations and are effectively earthquake-resistant [78]. Furthermore, existing challenges in Bangladeshi ready-made garments can be overcome by exploring green human resource management that supports the SGDs [79,80]. Nevertheless, the implementation of sustainable water management policy supports the food industry, which in turn supports the fulfilment of SDGs in Bangladesh [81–83].

## 6. Study Findings with Respect to Policy and Execution Status of CE and SDGs in Bangladesh

This section discusses the summary of the study findings with respect to the relevance of the concepts of CE practices to the UN's SDGs and the policy initiative of the Bangladeshi government to fulfil SDGs with respect to CE practices. In the study, it is realised that not

all but several SDGs are closely related to CE practices that need special attention to fulfil the target efficiently. For instance, SDG 12 (sustainable consumption and production) is directly related to CE practices, while SDG 14 (oceans) is not directly linked to CE. It can be also mentioned that the principles and practices of CE are transversal, meaning that CE practices would be necessary to implement many SDGs, and vice versa.

The findings from the study recommended that CE practices can support the achievement of several SDG targets directly, such as SDG 6 (clean water and sanitation), SDG 7 (affordable and clean energy), SDG 8 (decent work and economic growth), SDG 11 (sustainable cities), SDG 12 (sustainable consumption and production) and SDG 15 (life on land). It is also noticed that other SDGs such as SDG 1 (eliminating poverty), SDG 2 (ending hunger and sustainable food production), SDG 13 (climate change), SDG 14 (oceans), etc., are also indirectly supported by CE practices.

From the study, it is also realised that although CE helps in achieving SDGs targets, at the same time, SDGs also promote CE practices. There are many SDGs that are not relevant to CE but promote CE practices. For instance, SDG 4 (quality education) and SDG 9 (industry, innovation and infrastructure) indirectly facilitate CE practices. From these findings, it is recommended to further study, with more empirical evidence, the dependency between CE and the SDGs. This outcome answers the first research question, which is related to the relevance between CE practices and the UN's SDGs. To answer this research question, this study adopted a rigorous literature survey to extract the relevant information related to the interdependencies of CE and SDGs.

With respect to the second research question, the research studied how the Bangladeshi government has taken various strategic policies and guidelines to fulfil the UN's SGDs with respect to CE practices. Bangladesh is characterised as a traditional society. The society is less conscious of CE practices, in terms of awareness, attitude and mind-set, as reflected by the recycling rate disparity. The key issues encountered in the country are domestic and industrial wastes disposal and plastic clog, especially in urban areas, making the country highly ranked in terms of environmental hazards and concerns. Figure 2 visualises an example of solid waste generation in Bangladesh [84]. From Figure 2, it is observed that the waste generation per capita in urban areas is 0.41 kg/day and the waste collection efficiency is 44.30% to 76.47 % of various waste such as municipal waste, industrial waste, agricultural waste, medical waste and e-waste. From this study, it is noticed that the waste collection rate needs to be increased as much as 100%, if possible, to respect environmental sustainability in Bangladesh.

Figure 3 displays a study conducted in 2019 related to plastic waste generation in urban areas in Bangladesh. It shows that out of 821,250 tonnes/year generated plastic wastes in Bangladesh, 36% goes to the recycling industry, while 39% goes to landfills and the remaining 25% is left as leakage or unattended, of which a substantial amount finds its way into marine environments [85].

Furthermore, Figure 4 displays a study that is conducted in 2017 on the life cycle of plastic products from waste to recycling. From the figure, it is revealed that collected plastic products from various sources (e.g., industrial, household, agriculture, office, etc.) are processed through different methods such as recycling, landfills and energy conversion [86]. Some branded business companies in Bangladesh make most of the waste with single-use plastic materials, as shown in the figure. A private initiatives and incentives are there for used recyclable materials. These limited efforts do not facilitate and attract wider involvement in the issue of CE development. Additionally, the regulations and logistic facilities on fast consumer goods are almost insufficient.

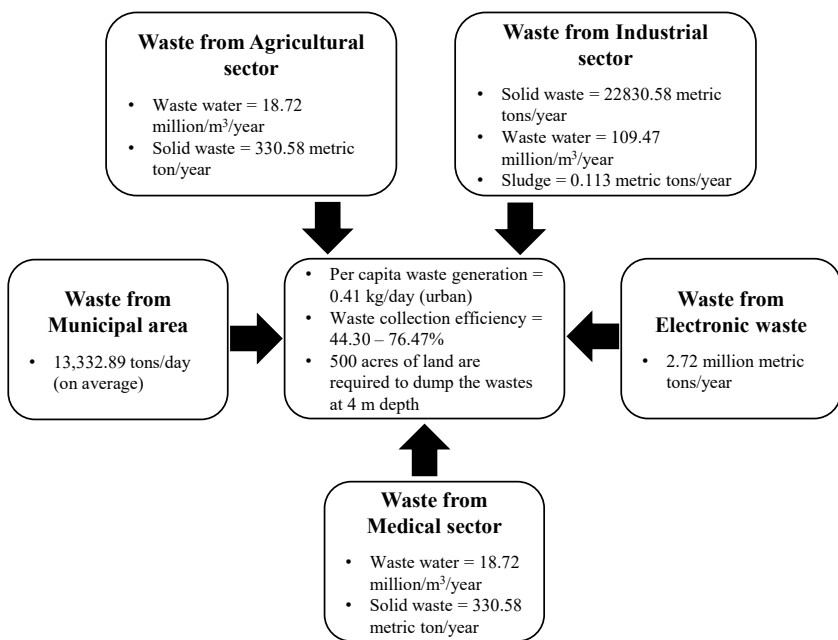

**Figure 2.** Solid waste generation in Bangladesh (adapted from [83]).

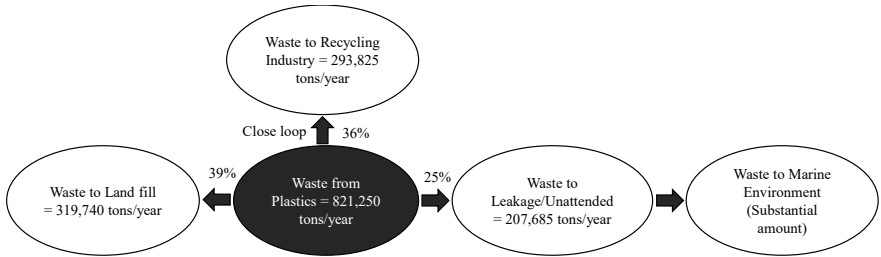

**Figure 3.** Plastic waste situation of urban areas of Bangladesh (adapted from [84]).

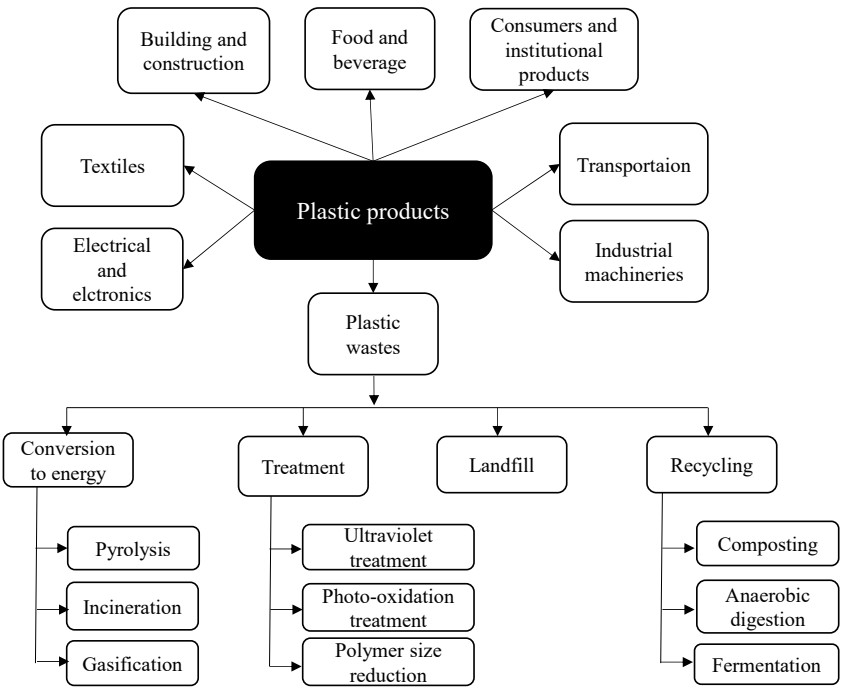

**Figure 4.** Display of plastic products life cycle from waste to recycling (adapted from [84]).

This study finds that Bangladesh is limited in administrative, land and financial capacity in executing the policies and programmes to fulfil SDGs. It has been observed that the national government has made "inadequate efforts to provide adequate supplementary policies" to address the limitations. The government of Bangladesh has used CE as a tool that targets SDG achievement through national policy guidelines, stakeholder involvement and business interests in the country. The research finds that there has been a misalignment between government policies and practices for the implementation of the global agenda. Thus, the progress is mixed, and the achievement of the UN's SDGs has been at stake [27,87].

### 6.1. Limited Public Awareness and Inadequate Support Facilities

The national government of Bangladesh is the motivator for CE development with legislations and regulations; however, the public has less involvement in CE principles and practices. It is noticed that government legislations and regulations have not aroused public participation. Thus, the notion of CE remains insufficiently understood by the general public [21]. The general public still has a conservative mind-set of one-use and disposal as the major way to treat waste. They are not yet familiar with the concept of recycling and reuse, which is a hindrance to CE development in Bangladesh.

The Bangladeshi government is about to ban the manufacturing of environmentally harmful material-based construction materials, such as fire clay bricks, which are made in brick kilns after burning coal and forest woods as fuel and damage the environment [88,89]. The construction industry in Bangladesh is yet to adopt alternative technology for building materials with locally available raw materials, such as eco-bricks, which is a cleaner brick made from burn free-technology [88,89]. The "recycled construction materials are not in use widely in the country. As the public have little awareness about the consequence of the harmful materials, the government and businesses are not under pressure to "implement the related policies widely for ecology and circularity in the country, which is also seen in other developing countries.

The government system in Bangladesh is comparatively less modernised in practice. The facilities to support respective policies are mostly sedentary. It has been noticed that the universal exemption standard and monitoring is "almost absent in practice, which hinders its effectiveness. Although government legislation supports the establishment of material recovery facilities, the actual implementation is, as it was claimed, unsatisfying. The implementation status is not consistent amongst rural areas, cities and towns. The green programme balances the value of cost, quality and environmental impacts during the process of product and service procurement; however, the reflected outcome is not as expected. "It requires a comprehensive approach to integrating the green agenda with the long-term development agenda of the country as a whole" [67]. It is understood that the effectiveness of governmental strategies execution in Bangladesh is weak, and, therefore, a hindrance to local CE development in the country.

Logistical support and sophisticated cost-effective technology are vital for circularity. It has been understood that Bangladesh is lagging in logistical support and technological development for CE due to insufficient funding. Thus, optimisation of CE practice in the country is limited in garbage disposal with poor efficiency, without effective reduction and replacement. The financial support from the plastic recovery system fee, for instance, is not in operation practically that could contribute largely to CE development in the country. This hinders governance capacities and operational capabilities in innovative technological development, and thus, the country is yet to demonstrate CE development. Innovative and cost-effective novel technologies can increase national and local CE progression, improving environmental conditions significantly, but are almost absent in Bangladesh.

### 6.2. Misalignment between Policy and Execution

Although Bangladesh as a developing country has made considerable progress, it faces many challenges. The green development issues in the country do not have a

legally binding and enabling framework as they do in Korea [67]. The strategies and policies as a whole are free-standing and focused narrowly on some specific sectors, such as forestry, agriculture, water and industry, not specifically on CE development. They are less integrated and well-linked to national development strategies, left somewhat vague without following implementable action plans. A wide-ranging policy framework involving fiscal, trade, infrastructure, pricing for public utilities, public investment, water management, industrial policies, agricultural, etc., that facilitates private investment, production, export and import decisions is functionally absent in the country [67].

Bangladesh is a tropical country where natural disasters, flooding and water logging are common. This is exacerbated during the monsoon period. Irregularity in the waste disposal and plastic clog hinders water diversion, regularly causing flooding. This creates environmental and social problems simultaneously, affecting citizens' quality of life. The strict monitoring of waste disposal regulation and plastic bag banning strategies might be useful, reducing pollutants from waste products and waste disposal. Environmental problems, such as serious pollution, drainage blockage and plastic clogs, oppose the thorough development of CE. CE can address the root cause of all these problems, reducing conventional plastic products and related disposal. Bangladesh has addressed environmental issues, such as climate change, pollution and hazards brought by the production and existence of plastic solid waste, which may relate to CE development. It has been realised that they exist in theory, but not properly executed in practice. The Bangladeshi government has implemented several environmentally friendly regulations that can promote CE and resolve SGD issues efficiently. For instance, the Bangladeshi government has imposed plastic bag banning, which intends to reduce plastic polythene products in the country. These polythene products are one of the main sources in the waste stream in the country's urban areas; however, the lack of proper monitoring is responsible for limited success. Moreover, the government has also undertaken necessary efforts to create environmentally friendly policies to advocate CE through increased local need and international pressure. The aims of these policies are to bring about liveliness, prosperity and healthy living in the country and economy. If the policies are properly monitored and managed, these policy initiatives could assist the country to realise the SDGs target, moving from the basic level of mere waste disposal to reach higher levels of circularity. Such initiatives also clearly address the second research question regarding the Bangladeshi government's efforts to fulfil the UN's SDG targets.

## 7. Conclusions, Implications and Future Research Directions

This study provides a qualitative answer to the research questions related to the CE and UN SDGs. It focuses on national government policy responses and practices regarding CE and SDGs, with a focus on Bangladesh. There are varying degrees of momentum with CE development in the national policy response to SDGs. This paper is one of many contributions to the discussion on CE and SDGs, contributing to the academic literature in the area of sustainability and the circular economy in relation to a developing country, Bangladesh, and its experience with pursuing the SDGs. It initiates academic debate on sustainable development agendas within the context of developing countries. It has intermingled the theoretical debates of sustainability into sustainable development activities with CE practices in Bangladesh. Ultimately, the effectiveness of Bangladeshi national legislation can affect its social, economic, environmental and developmental issues, which in turn affect the global sustainable development agenda. However, the final outline of a CE in the country is still a work in progress, and there is much to explore in terms of potential and prospects.

The study findings have implications for national governments in developing countries worldwide with similar social, cultural and economic backgrounds. The UN's SDGs may take practical lessons for the reassessment of their policies and programmes in order to adjust to the local situation for active development partnerships with developing countries. Considering the impact of the current COVID-19 pandemic crisis on sustainable

development, a contextual CE model is an eminent one that can be replicated with differentiation. Although the Bangladeshi government has taken initiatives for SDG realisation, it should adopt policies and programmes specifying CE principles and practices that are practically applicable for the country. A wider involvement of government, public and private institutions as well as voluntary sectors of society may accelerate the implementation. As CE is a global trend, the research suggests that broad, conscientious connection and collaboration at the national level are essential. More emphasis should be placed on logistical and managerial capacity to handle the situation timely.

More research into value chains and the discipline of resource and energy economics could make the CE a reality both in Bangladesh and abroad in the near future. Future research should consider specific practices and factors, reviewing them regularly to monitor the progress and underlying contextual factors of social and economic aspects of a particular country. The current pandemic put the world's overall situation in a state of devastation. As national governments are the main agents for the implementation of the global development agenda, they need to come forward with more practical efforts for the "new normal" in terms of the present social and economic situation, using the approach to "think globally and act locally" in the time of the pandemic. The very nature of CE can help to improve the existing fragile global socio-economy through the circularity of existing limited resources.

This research study was conducted with a focus on a specific developing country, Bangladesh. It is hard to overtly generalise the findings from this study to other countries, but the findings are helpful to encourage deeper insights and further rigorous research on the issues of CE and SDGs. Further research on the policy stakeholders and the plight to promote local CE-related business initiatives might be an option. Additionally, a potential future research direction could evaluate the impact of COVID-19 and emphasise CE development in developing countries. Moreover, in the future, studies can be adopted with an enhanced quantitative approach by collecting more data from various arms of the Bangladeshi government responsible for CE and SDGs. Moreover, further in-depth research on the discussed issues with cross-country comparisons can indicate the appropriate development directions that can help facilitate the achievement of SDG targets by 2030.

There are several limitations to this study that should be mentioned. First of all, the sample size for the interviews was not high enough due to several restrictions imposed by the current COVID-19 situation. It was not possible to conduct face-to-face interviews, and telephone calls may not be efficient enough to collect the specific viewpoints related to CE and SDGs from the experts. Secondly, it was difficult to acquire information related to Bangladeshi government policies on CE and SDGs. Moreover, there were no funding sources to conduct this study. Despite such limitations, the authors tried hard to make this study as standard as possible and provide high-quality, valuable insights.

**Author Contributions:** The research was designed and performed by M.A. and A.S. The data were collected and analyzed by M.A. The paper was written by M.A. and A.S. and finally checked and revised by S.P. All authors have read and agreed to the published version of the manuscript.

**Funding:** This research received no external funding.

**Institutional Review Board Statement:** Not applicable.

**Informed Consent Statement:** Not applicable.

**Data Availability Statement:** Not applicable.

**Conflicts of Interest:** The authors declare no conflict of interest.

## Appendix A. Key Questions That Were Asked during the Telephone Interview Discussion

(a)    What do you know about the relationship of SDGs and circularity/circular economy?

(b)     What are the key legislations, policies, regulations, etc., related to circularity in Bangladesh?

(c)     How does the CE promote the achievement of SDG targets?

(d)     How do Bangladeshi government initiatives support the fulfilment of SDGs?

(e)     Do you think the business of recycling, circularity, etc., is long-term and profitable? If so, how?

(f)     What are your suggestions/advice to the Bangladeshi government regarding the circular economy?

(g)     What are the limitations of not addressing CE properly to fulfil SDGs?

(h)     What are future means to fulfil SDGs through CE practices in Bangladesh?

(i)     What are your specific suggestions for the Bangladeshi government to achieve the SDGs?

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
