# Peer review of "Influence of Circular Economy Phenomenon to Fulfil Global Sustainable Development Goal: Perspective from Bangladesh"

_sustainability, doi:10.3390/su132011455_

Round 1

Reviewer 1 Report

Thank you for giving me the opportunity to read your paper ”Influence of Circular Economy Phenomenon to Fulfill Global Sustainable Development Goal: Perspective from Bangladesh.” You addressed a very important topic in your manuscript, focusing on the influence of the circular economy phenomenon in Bangladesh. However, I also have some concerns concerning your work, which I will detail in the following.

Introduction comment. In the introduction, it should be mentioned how you conducted your study and your study context. The economic context is not described when the author(s) made the introduction part. Outlines the content of the paper – its aims and defines key terms. Take into account to move subsection 1.1 Study Aims and Research Questions, and insert here You state research questions at the beginning of your paper but do not explicitly link to the literature review/Theoretical Narratives. The basic ideas should be introduced in the research questions in the introduction and could enrich them.

Theoretical Narratives comment.

Restructure this part and insert ideas from parts 3.1., 4, and 5. We would expect to have more research questions at the end of the comprehensive literature review. Connect these research questions to your conceptual framework. What would answering these questions tell us that we don`t already know/ How do these questions form a coherent set that will guide your study?

Methodological Consideration comment

For the description of the methodology, I suggest that you reconceptualize your paper based on the descriptions depicted in accepted journal articles. At the moment, it is somewhat unclear how you conducted your study. What does ‘unstructured interviews via phone’ (r.230) really mean? It is not appropriate to just say, “Utilizing previously understudied archives including personal knowledge, correspondence, and other documents, this article highlights key concepts and discussion points, and accordingly conclusions” (rr. 224-226). 

Unfortunately, without a proper methodology chapter, it is difficult to trust the results of your paper as the reader does not have proof to evaluate the reliability and validity of the research. 

Findings and Discussion comment.

Take into consideration to structure of your results section around research questions that emerged from your analysis of the data. For each research question, make observations about what the data showed. You must clarify and support these observations with direct quotations and report relevant information: What do the results mean? Why do the results matter? Full transcripts of your interviews can be included in an appendix. I think that pictures are not relevant for an academic manuscript.

Conclusions, Implications and Future Research Direction comment.

First, clearly state the answer to the main research questions. Then answer these questions: What can`t the results tell us? What practical actions or scientific studies should follow?

I hope these comments are helpful when revising the manuscript, and I wish you all the best for your future research in this area.

Author Response

Please have the responses in the attached file

Reviewer 2 Report

Thanks for this paper and the topic. The paper addresses a nice topic but lacks of strong research methodology. I have comments below.

The research methodology is very poor. The authors must explain better, especially in section 3. For example, is an unstructured interview relevant? 

And for the findings, what can other countries learn from this paper?

Author Response

Please have the responses in the attached file.

Reviewer 3 Report

Review Report

The manuscript reviews the current circular economy practice and its influence on achieving the UN's sustainable development goals (SDGs) in Bangladesh. The authors have appropriately identified the research question and tried to answer these questions with theoretical consensus. The manuscript has been structured well; however, the content can be improved both quantitatively and qualitatively.

The section-specific comments can be seen as follows;

The initial part of the abstract is well structured. The authors indicate in the abstract that 'the manuscript's findings can be implicated for policy and program reassessment.' However, discussion regarding the implication of such suggestions is missing in the main manuscript. The main manuscript also misses the discussion regarding how the suggestive solutions can help in the COVID-19 pandemic crisis.

 The introduction part is well organized. However, discussion on environmental sustainability is lacking (Most of the discussion is around social and economic aspects of sustainability). In the introduction section, the authors have mentioned that "the manuscript advances with a bird's eye view on international expectations, national efforts, and recommendations and implications for the policy and execution." The former part is explained in detail; however, not sure whether the study covers the latter part of the recommendation and implication for policy and execution. The discussion of these points will improve the manuscript significantly.  

In the theoretical narrative section, definitions like sustainability, sustainable development, etc., are not encouraged. Instead, focus can be given to seven different 'R' in the circular economy concept. There is a repetition of some concepts which can be easily avoided. The manuscript has mentioned that 'the current economic activities pose a serious threat to sustainability; however, the justification is missing. The steps for transitioning to circularity is well explained, and the entire section can be rebuilt around this. How are innovative business models always profitable? A justification would be beneficial.

In SDGs and CE section, goal no. 6 is regarding access to water and sanitation. The authors have mentioned that goal no. 6 corresponds to energy, which needs to be corrected.  There is some repetition of CE principles, which can be avoided easily. Figure 1 corresponds to the conceptual framework of CE for SDG's. However, the addition of a bridge between CE and policies can be a plus point.

In the methodology section, the authors have mentioned that they have arranged some six unstructured interviews. It would be significant if authors can share the questionnaires in tabular form. This section can be explicitly modified with questionnaires, datasheets, etc., as it will give significance to the manuscript.

In B'Desh and CE sections, the authors have mentioned several policies which the B'Desh government is implementing. This section gives an essential dimension to the present manuscript. Moreover, a table can be prepared with all the policies, their description, and potential implementation. The subsequent section (B'Desh and SDGs) has the potential to improve.  

Only one case is discussed in the finding and discussion section (though the data was collected from the six policymakers via unstructured interview). The year of the study is not mentioned in figure 2. In figure 3, the numbers do not match up with the percentage share. The authors have mentioned that the government is about to ban environmentally harmful material-based construction materials. Can authors describe those materials and their counterparts? (Materials which can be used to replace such environmentally harmful materials). Can the authors explain the sentence "the government and business companies are not under due pressure for ecology and circularity?" How will public awareness (or lack of it) impact policymaking? A little clarification is expected. The authors have mentioned that the country does not have appropriate guidelines regarding disposal and recycling; however, the earlier section mentions various schemes/ policies to promote CE. Can authors restructure this discussion? What happens to the waste after the trash-to-cash scheme? Can the authors explain this with a suitable example? Some of the legends in figure 4 are not visible. In the paragraph starting from lines no 374-384, the authors have concluded that "the financial support is not in operation that supports the CE"; however, on page 6 (line 266), the authors have mentioned that "B'Desh government has introduced the mix policy instruments, such as fiscal incentive…." Such contradictions need to be avoided. The emphasis can be given to the potential solution of the existing issues.

In the misalignment between the policy and execution section,  the authors have focused on forestry, agriculture, water, and industry sectors. The authors should also discuss the other remaining sectors, which have been discussed earlier. Some contradictory statements need to be modified, as mentioned earlier. The authors have mentioned that the "B'Desh gov. has illustrated the environmental friendly regulations to promote CE", can authors prepare a table and mention all the policies and regulations that might help in mitigating the issue? This section can be improved quantitatively by discussing more examples.

In the conclusion section, the authors have mentioned that more emphasis should be given to logistic capacity… but this part has not been discussed explicitly in the manuscript.

Reference no. 25 is the Brundtland Report published in 1987, not in 1988. Kindly check the publication date.  

Overall, the structure of the manuscript is good, but the content can be specifically modified. The sentence restructuring is required in some parts. Some words, like dearth, tantamount, infers, regraded grapple, etc., can be simplified. Kindly focus on the choice of the words.

Author Response

Please have the responses at the attached file.

Reviewer 4 Report

Manuscript ID: sustainability-1348786

Title: "Influence of Circular Economy Phenomenon to Fulfill Global Sustainable Development Goal: Perspective from Bangladesh"

FIRST REVIEW REPORT

In the paper, the Authors highlighted the relationships between Circular Economy practices for the implementation of the Sustainable Development Goals.

The topic of the paper is interesting as well as the potential academic contribution of the work, but the Authors should improve the research according to the following indications.

1. In the introduction section, the Authors should discuss more international situation, regulations, and approaches, and should motivate the research to be of high interest for the addressees.

2. Section 7. is too brief and should be extended. In particular:

- The conclusions should be expanded.

- Policy implications should be expanded and implication for the theory and practice should be considered.

- Limitations of the study should be considered.

3. The following studied should be considered:

https://doi.org/10.3390/su12125120

https://doi.org/10.17221/343/2020-AGRICECON

4. All tables and pictures should report their sources.

5. An extensive editing of English language and style is required. 

Author Response

(The authors gave the same response as above.)

Reviewer 5 Report

  • a greater clarity of arguments is needed in the different sections.
  • the theoretical framework should be improved

Author Response

Comments and Suggestions for Authors
  • a greater clarity of arguments is needed in the different sections.
  • the theoretical framework should be improved

Response 1: OK. Thanks a lot. The alignment of the manuscript is checked all the way and revised accordingly. Theoretical framework is improved further too.

Round 2

Reviewer 1 Report

The methodological rigor has major problems, and the issues raised in my comments didn`t satisfactorily addressed in the revised manuscript.

Author Response

Response to Reviewer 1 Comments

Point 1: The methodological rigor has major problems, and the issues raised in my comments didn`t satisfactorily addressed in the revised manuscript.

Response 1: OK. Thanks a lot. Study methodology section is revised and improved all the way accordingly. Section 3.1 and 3.2 is newly added and section 3.3 is improved further. Please see all the changes as highlighted with yellow colour.

Reviewer 2 Report

The research methodology has not been improved well. I suggested the authors read some papers using the same methods. Some examples questions in the research methodology section:

  • How did you conduct the literature review?
  • How did you have results from the Delphi approach?
  • How did you select experts?

Author Response

Response to Reviewer 2 Comments

The research methodology has not been improved well. I suggested the authors read some papers using the same methods. Some examples questions in the research methodology section:

  • How did you conduct the literature review?
  • How did you have results from the Delphi approach?
  • How did you select experts?

Response 2: OK. Thanks a lot. Study methodology section is revised and improved all the way following your three questions accordingly. Section 3.1 and 3.2 is newly added and section 3.3 is improved further by addressing how we have received results from the Delphi method and how the experts were selected. Please see all the changes as highlighted with yellow colour.

Reviewer 3 Report

The authors have sufficiently modified the manuscript according to the suggestions.

The standard of the manuscript in its present form is sufficiently higher to get it accepted in the journal.

Few minor corrections to consider before the final acceptance;

  1. Manuscript alignments need to be checked.
  2. In table 1, educational qualifications and age are not desired. I believe their relative experience is most important.
  3. In figure 3, values are still not changed (which doesn't correspond to the percentage share).

Author Response

Response to Reviewer 3 Comments

Point 1: The authors have sufficiently modified the manuscript according to the suggestions.

The standard of the manuscript in its present form is sufficiently higher to get it accepted in the journal.

Few minor corrections to consider before the final acceptance;

  1. Manuscript alignments need to be checked.
  2. In table 1, educational qualifications and age are not desired. I believe their relative experience is most important.
  3. In figure 3, values are still not changed (which doesn't correspond to the percentage share).

Response 1: OK. Thanks a lot. The alignment of the manuscript is checked all the way and revised accordingly. Table 1 also updated by adding experiences of the experts. In Figure 3, the percentages are rechecked and it seems ok (36% for recycling industry + 39% for land fill +25% for leakage/unattended =100%).

Reviewer 4 Report

Manuscript ID: sustainability-1348786

Title: "Influence of Circular Economy Phenomenon to Fulfill Global Sustainable Development Goal: Perspective from Bangladesh"

SECOND REVIEW REPORT

The Authors improved their manuscript according to the suggestions of my previous review report.

Now the paper is suitable for the journal.

Contratulations!

Author Response

Response to Reviewer 4 Comments

Point 1: The Authors improved their manuscript according to the suggestions of my previous review report.

Now the paper is suitable for the journal.

Contratulations!

Response 1: OK. Thanks a lot.

Reviewer 5 Report

I accept this version of the article

Round 3

Reviewer 1 Report

Article has serious flaws, additional experiments needed, research not conducted correctly.

Reviewer 2 Report

Thanks for the revision.